# Optimization of the Decolorization of the Reactive Black 5 by a Laccase-like Active Cell-Free Supernatant from *Coriolopsis gallica*

**DOI:** 10.3390/microorganisms10061137

**Published:** 2022-05-31

**Authors:** Amal Ben Ayed, Bilel Hadrich, Giuliano Sciara, Anne Lomascolo, Emmanuel Bertrand, Craig B. Faulds, Héla Zouari-Mechichi, Eric Record, Tahar Mechichi

**Affiliations:** 1Laboratory of Biochemistry and Enzymatic Engineering of Lipases, Ecole Nationale d’Ingénieurs de Sfax (ENIS), University of Sfax, Sfax 3038, Tunisia; hela.zouari@isbs.usf.tn; 2UMR1163, Biodiversité et Biotechnologie Fongiques, Aix-Marseille Université, INRAE, 13288 Marseille, France; giuliano.sciara@inrae.fr (G.S.); anne.lomascolo@univ-amu.fr (A.L.); emmanuel.bertrand@univ-amu.fr (E.B.); craig.faulds@univ-amu.fr (C.B.F.); eric.record@inrae.fr (E.R.); 3Laboratory of Enzyme Engineering and Microbiology, Ecole Nationale d’Ingénieurs de Sfax (ENIS), University of Sfax, Sfax 3038, Tunisia; bilel.hadrich@enis.tn

**Keywords:** RB5 decolorization, *Coriolopsis gallica*, laccase-like activity, optimization, response surface methodology

## Abstract

The textile industry generates huge volumes of colored wastewater that require multiple treatments to remove persistent toxic and carcinogenic dyes. Here we studied the decolorization of a recalcitrant azo dye, Reactive Black 5, using laccase-like active cell-free supernatant from *Coriolopsis gallica*. Decolorization was optimized in a 1 mL reaction mixture using the response surface methodology (RSM) to test the influence of five variables, i.e., laccase-like activity, dye concentration, redox mediator (HBT) concentration, pH, and temperature, on dye decolorization. Statistical tests were used to determine regression coefficients and the quality of the models used, as well as significant factors and/or factor interactions. Maximum decolorization was achieved at 120 min (82 ± 0.6%) with the optimized protocol, i.e., laccase-like activity at 0.5 U mL^−1^, dye at 25 mg L^−1^, HBT at 4.5 mM, pH at 4.2 and temperature at 55 °C. The model proved significant (ANOVA test with *p* < 0.001): coefficient of determination (R²) was 89.78%, adjusted coefficient of determination (R²_A_) was 87.85%, and root mean square error (RMSE) was 10.48%. The reaction conditions yielding maximum decolorization were tested in a larger volume of 500 mL reaction mixture. Under these conditions, the decolorization rate reached 77.6 ± 0.4%, which was in good agreement with the value found on the 1 mL scale. RB5 decolorization was further evaluated using the UV-visible spectra of the treated and untreated dyes.

## 1. Introduction

Water is essential for all forms of life, making it the most important resource on Earth. However, water decline and depletion is an escalating problem driven by the increasing global population [1,2]. This increase in population also increases pollution, regardless of whether wastewater streams come from domestic, industrial, agricultural, or other origins [3]. A telling example is the use of pesticide-contaminated water for plant irrigation, which leads to the spread of carcinogenic chemicals in soil and, successively, all along the food chain [4].

The textile industry is the oldest sector in the world’s economy [5] and is still the world’s largest industry, employing huge numbers of workers worldwide (1.5 million in Brazil and 8 million in China) and producing huge amounts of textiles (1.3 million tons in Brazil, and 79.29 million tons in China) [6,7,8]. The sector continues to grow with the growing global population, increasingly productive machinery, and a wider variety of colors [9]. The textile industry uses a large number of dyes that are classified according to their chemical structure (chemical classification) or application method (tinctorial classification) [10,11]. Most of the dyes used today are synthetic unsaturated organic or aromatic compounds, which have a complex structure and are applied in a range of domains from cosmetics and pharmaceuticals to paper and food [12,13]. The best-known class of synthetic dyes are azo dyes, which possess an azo linkage (-N=N-) close to the aromatic rings that confer their high resistance to oxidizing agents and photocatalytic stability [14,15]. Reactive Black 5 (RB5) is a non-biodegradable azo dye known to be highly resistant to chemicals and light [13,16], and it is one of the more common reactive dyes used in the textile industry [17]. Textile industries employ three major processes, i.e., washing, dyeing, and finishing, that together use 50 to 100 L of water per kg of wet fiber produced [18,19]. These textile-industry processes also produce effluents that are loaded with toxic and carcinogenic chemicals and dyes (0.01 to 7 g L^−1^ of dyes) [20,21]. These effluents cause extensive water pollution and thus severe problems for fauna and flora as well as toxicity for humans [22,23,24]. Therefore, this alarming situation necessitates efforts to treat these effluents before they are released back into the environment. The textile industry faces a major challenge to address this damage and the allied water pollution crisis [19].

Wastewater treatment spans a series of processes designed to remove organic and inorganic pollutants from water before it is reused or returned to the natural environment. These wastewater treatments can employ chemical, physical and/or biological treatment processes in various combinations and configurations [24,25]. Physical unit processes include filtration, coagulation/flocculation, and ion exchange. Chemical treatment processes revolve around chemical reduction, advanced chemical oxidation, and electrolysis. Biological unit processes typically include aerobic, anaerobic or combined processes [26,27,28]. Physicochemical treatments have proven expensive and relatively inefficient as they fail to curb the spread of dangerous chemicals and lead to potential ecological risks [28]. Consequently, biological approaches emerge as the best candidate solutions to remove dyes and preserve clean water resources [27,29]. Biological processes use the enzymatic machinery of bacteria or fungi as their main tools, which make them non-toxic, eco-friendly, and low-energy solutions [30].

White-rot fungi are the most intensively-studied heterogeneous organisms in the literature for their ability to degrade lignin and other aromatic components, in addition to their potential application in the textile industry to remove dyes from wastewater. Research has pursued white-rot fungi mainly because they mobilize an extracellular enzyme system composed of peroxidases and laccases [31,32]. For instance, *Pycnoporus sanguineus*, *P. coccineus*, *Trametes versicolor*, *Trametes hirsuta*, and *Phanerochaete chrysosporium,* are fungal strains that widely show efficiency for decolorizing textile dyes [31,33,34,35].

*Coriolopsis gallica* is an example of a white-rot basidiomycete that produces high levels of extracellular enzymes such as laccase [36]. Studies have reported the ability of *Coriolopsis gallica* to degrade dyes such as Remazol Brilliant Blue R (RBBR), an anthraquinone dye [37], Reactive Blue 198 [38], an azo dye, and Lanaset Grey G, a metal textile dye [39], as well as phenolic compounds [40] and polycyclic aromatic compounds such as carbazole, dibenzothiophene, and dibenzofuran [41].

Laccases (benzenediol, oxidoreductase, EC 1.10.3.2) are multi-copper enzymes that catalyze the oxidation of various substrates, such as phenols, aromatic thiols, aniline, and some inorganic ions via the reduction of O_2_ to H_2_O [42,43]. These enzymes are found in plants [44], bacteria [45], insects [46], and fungi [43,47]. Fungal laccases, especially those from white-rot fungi, are among the best-described and widely-used enzymes in various biotechnological applications from pharmaceuticals [48] to cosmetics [49], pulp bleaches [50], food [51], bio-detergents [52,53] and textile dye decolorization [54,55,56]. However, laccases are unable to directly oxidize complex compounds that have a higher redox potential than laccases or steric hindrance preventing access to the catalytic site of the enzyme [57,58]. These cases demand the use of low-molecular-weight redox mediators that act as electron shuttles to help oxidize non-phenolic compounds [59,60]. There are two broad classes of redox mediators: synthetic and natural [61]. Among the synthetic redox mediators, 1-hydroxybenzotriazole (HBT) is a well-known N-OH-type mediator [59]. Other examples of N-OH-type mediators include *N*-hydroxyphthalimide (HPI), violuric acid (VIO), *N*-hydroxyacetanilide (NHA), 2,2,6,6-tetramethylpiperidin-1-yloxyl (TEMPO), and 10-(3-dimethylaminopropyl) phenothiazine (Promazine) [59,62]. Natural mediators include syringaldehyde, vanillin, acetosyringone, acetovanillone, and *p*-coumaric acid [62].

This study aims to evaluate the potential of a crude *Coriolopsis gallica* laccase for the decolorization of the recalcitrant azo dye RB5. The fungal strain used in this study had been isolated from decayed acacia wood in northwest Tunisia [37]. Statistical optimization was performed using Response Surface Methodology (RSM) (coupled screening and Box–Behnken designs) to determine optimized conditions for multifactorial experimentation and the interactions between variables. The optimized conditions for decolorization were first verified at a small scale (1 mL) and then tested in a larger volume of 500-mL. In addition, the UV-visible spectra of treated and untreated dye were analyzed.

## 2. Materials and Methods

### 2.1. Fungal Strain and Culture Conditions

This study used *Coriolopsis gallica* strain CLBE55 [ON340792] for laccase production. *Coriolopsis gallica* strain CLBE55 was deposited at the culture collection “Centre International de Ressources Microbiennes” (CIRM) under accession number BRFM 3473.

Solid cultures of *C*. *gallica* were performed on PDA media that contained 39 g of dehydrated media (Accumix) suspended in 1000 mL of distilled water sterilized by autoclaving at 120 °C for 30 min. Liquid pre-cultures were performed in 25 mL of Malt Extract medium (Sigma–Aldrich, St. Louis, MI, USA) containing 30 g Malt Extract per L at pH 5.5 sterilized by autoclaving at 120 °C for 30 min. The pre-cultures were inoculated with 3 agar plugs (6 mm in diameter) cut from the growing edge of a plate stock culture and incubated at 30 °C for 3 days at 160 rpm. Mycelia from these three-day precultures were then partially ground down using glass beads (0.6 mm). The mycelial mixture obtained was used to inoculate 500-mL Erlenmeyer flasks containing 100 mL of M7 medium [27,63]. The basal medium contained (in g L^−1^) glucose 10, peptone 5, yeast extract 1, ammonium tartrate 2, KH_2_PO_4_ 1, MgSO_4_, 7H_2_O 0.5, KCl 0.5, and 1 mL of trace element solution. Composition of the trace-element solution was (in g L^−1^): B_4_O_7_Na_2_, 10H_2_O 0.1, CuSO_4_, 5H_2_O 0.01, FeSO_4_, 7H_2_O 0.05, MnSO_4_, 7H_2_O 0.01, ZnSO_4_, 7H_2_O 0.07, (NH_4_)_6_Mo_7_O_24_, 4H_2_O 0.01. pH was adjusted to 5.5, and cultures were incubated in a rotary shaker for 7 days at 30 °C and 160 rpm. On day 3 of incubation, 300 µM CuSO_4_ was added as a laccase inducer.

### 2.2. Laccase-like Activity Assays of the Cell-Free Supernatant from Coriolopsis gallica

The supernatant of a 7-day culture of *C. gallica* was filtered on a Miracloth membrane (Merck, Fontenay-sous-Bois, France) and centrifuged at 30 °C for 5 min at 10,000× *g* prior to use. The laccase-like activity of the cell-free supernatant was assayed by monitoring the oxidation of 5 mM 2,6-dimethoxyphenol (DMP) in 50 mM citrate buffer, pH 5 (469 nm, Ɛ_469_ = 27,500 M^−1^cm^−1^) in the presence of 50 µL of supernatant. The assay was carried out at 30 °C for 1 min. One unit of DMP-oxidizing activity was defined as the amount of enzyme oxidizing 1 µmol of substrate per minute.

### 2.3. Experimental Design and Data Analysis

An experimental design was used to optimize the enzymatic decolorization of the recalcitrant azo dye Reactive Black 5 (RB5) (Aldrich Chemical Co., St. Louis, MO, USA) by using the culture supernatant of *C. gallica*. The chemical properties of the RB5 dye are reported in Table 1.

### 2.4. Plackett–Burman Design

Plackett–Burman experimental designs are useful first-step screening designs for identifying the most significant factors and weeding out uninfluential factors for further experimentation [64]. Here we applied a 15-run Plackett–Burman design, with 3 replicates including the center points, on five independent factors to determine their influence on RB5 decolorization. The five factors were laccase-like activity (x_1_), initial dye concentration (x_2_), HBT concentration (x_3_), pH (x_4_), and temperature (x_5_). The center point tested the linearity of the experimental points (external versus center point. The experimental design adopted required three levels, i.e., low (coded −1), medium (coded 0), and high (coded +1) (Table 2). Table 3 reports the Plackett–Burman design used and the percentage decolorizations achieved at 120 min. The reaction took a total volume of 1 mL using 50 mM citrate buffer. All experiments were performed in triplicate. Results are presented as means ± standard deviation (Table 3). The first-order form of the equation adopted in this part of the study is [65] (Equation (1)):(1)$${\hat{y}=\beta }_{0}+\sum_{i}\beta_{i}\cdot x_{i}\left(i=1\ldots k\right)$$
where ŷ is the fitted response (% decolorization at 120 min), β_0_ and β_i_ are the intercept and linear coefficient of the model, respectively, x_i_ is level-coded factor variable, and k is number of factors.

Decolorization was measured every 30 min over a 2-h incubation period by tracking the decrease in absorbance of the RB5 dye (598 nm). Percentage decolorization was calculated using the following formula (Equation (2)):(2)$$\mathrm{Decolorization}\left(\%\right)=\frac{A_{i}{-A}_{t}}{A_{i}}\times 100$$
where A_i_ is initial absorbance of the dye at the maximum wavelength before incubation with the enzyme, and A_t_ is dye absorbance after 2 h of incubation. All experiments were done in a total volume of 1 mL reaction mixture in 50 mM citrate buffer.

### 2.5. Response Surface Methodology Using a Box–Behnken Design

The best condition for RB5 decolorization was determined using RSM based on a Box–Behnken design with influential variables obtained previously via the Plackett–Burman design. A total of 46 runs (Table 4) were performed in triplicate with the same 5 variables and factor levels as described above. Decolorization was adjusted using a second-order polynomial equation, and multiple regression was performed on the data to obtain an empirical template associated with the factors. The general form of the second-order polynomial equation (Equation (3)) is:(3)$${\hat{y}=\beta }_{0}+\sum_{i}\beta_{i}\cdot x_{i}+\sum_{i}\sum_{i\neq j}\beta_{\mathrm{ij}}\cdot x_{i}\cdot x_{j}+\sum_{i}\beta_{\mathrm{ii}}\cdot x_{i}^{2}\left(i,j=1,\ldots ,k\right)$$
where ŷ is the response (percent decolorization at 120 min), β_0_, β_i_, β_ij_ and β_ii_ are the model’s intercepts, linear, interactions and quadratic coefficients, respectively, x_i_ is level-coded factor variable, and k is 5 in our experimental setup.

### 2.6. Design and Statistical Analysis

Minitab 16 Statistical Software (Minitab Inc., State College, PA, USA) was used for the experimental designs and statistical analysis. Factor coefficients were determined using the least-squares method. Analysis of variance (ANOVA) and a Student’s *t*-test were used to identify the level of significance of both the fitted model and the factors and their interactions. The probability (*p* < 0.05) and high F-values (Fisher’s test) demonstrated that the model and factors were significant. The quality of the model was analyzed using a coefficient of determination (R²), adjusted coefficient of determination (R² adj), and root mean square error (RMSE).

### 2.7. Decolorization Assay in 500 mL Volume

The optimal conditions determined for RB5 decolorization in a reaction volume of 1 mL were then tested on a larger scale (500 mL). Experiments were performed in duplicate in 1 L Erlenmeyer flasks containing 500 mL of the following reaction mixture: 50 mM citrate buffer pH 4.2, 25 mg L^−1^ RB5 dye, 4.5 mM HBT, and 0.5 U mL^−1^ crude *C. gallica* laccase. The reaction mixture was incubated at 55 °C for two hours, and 1-mL samples were taken every 30 min over the 2-h incubation period. Percentage rates of decolorization were calculated following Equation (2).

### 2.8. RB5 Spectrum Analysis

The UV-visible spectrum of RB5 was analyzed to visualize its color and absorbance peak shift. Decolorization was carried out for 24 h in reference conditions described in Appendix A. After 24 h, UV-visible spectra were measured on a Genesys 50 UV-VIS spectrophotometer (Thermo Fisher Scientific, Waltham, MA, USA) running in the 200–800 nm wavelength range at 2-nm increment steps. 

## 3. Results

### 3.1. Screening Design

Preliminary experiments were performed to fix the levels of factors. With a view to potential scale-up for use in textile industry applications, reaction time was set at 2 h. Note that we ran tests at several reaction times (>2 h) and found no change in the final decolorization of the dye. In addition, a previous study had demonstrated that the main laccase of *C. gallica* was active and stable at acidic pH (4–7) for 24 h of incubation retaining from 40 to 80% of its initial activity and the same high temperature (up to 50–55 °C) as tested here [37] and that HBT was a powerful mediator for dye removal at up to 5 mM [37]. Based on these results, we thus performed the experimental design in these conditions.

Table 3 shows the experimental percentages of RB5 decolorization for all 15 runs tested (Plackett–Burman design). All response values (percentage decolorization) presented very low standard deviations (from 0.1 to 3.2% units of decolorization) (Table 3), thus demonstrating good repeatability of all the experiments done in the different conditions. The Pareto chart of the standardized effects (Appendix A) shows that 4 of the 5 variables had significant effect on RB5 decolorization (*t*_factor_ > *t*_student_ (*p* = 0.05) = 2.024; *p* < 0.05) and that enzyme concentration was the most effective variable (*p* < 0.001) compared to pH (*p* < 0.001), initial dye concentration (*p* < 0.001) and HBT concentration which was the least effective variable (*p* < 0.01). Temperature was found to be a non-significant variable but was kept for the second step of the RSM-based optimization protocol for two reasons: (1) its *p*-value (0.073) was almost ruled in as significant (i.e., very close to the 0.05 limit of significance); (2) temperature was found to be insignificant as the responses at extreme points (−1; 1) were very close to each other, but the center point was very significantly different to these values (Figure 1; *p* < 0.001).

Statistical results obtained from Minitab 16 gave the model coefficient, and the fitted form of the equation was then represented as follows (Equation (4)):ŷ = 18.36 + 15.61·x_1_ − 8.03·x_2_ + 5.71·x_3_ − 15.39·x_4_ + 3.84·x_5_
(4)

A positive-signed main effect of a factor indicates that using higher levels gave the most effective response whereas a negative-signed main effect indicates that using lower levels was more effective. Thus, according to Equation (4), laccase-like activity (x_1_), HBT (x_3_) concentrations, and temperature (x_5_) were positive-signed whereas dye concentration (x_2_) and pH (x_4_) were negative-signed. Moreover, the coefficient of determination (R²), the adjusted coefficient of determination (R²(adj)), and the root mean square error (RMSE) were 85.62%, 83.35%, and 12.5% (units of RB5 decolorization), respectively, thus demonstrating the goodness-of-fit of the adopted model (Equation (4)).

Figure 1 illustrates the main effects plot of the five tested variables, which shows that no linear correlation could be found between center points and extreme points. This nonlinearity, which was very significant (*p* < 0.001), indicated a second-order model was able to fit decolorization as a function of all factors. We consequently applied a second response-surface experimental design for the same variables in our protocol optimization step.

### 3.2. Box–Behnken Design

Response surface methodology based on a Box–Behnken design was performed to optimize (via a second-order polynomial model) the conditions of RB5 decolorization by crude laccase obtained from *C. gallica.* The Box–Behnken design also statistically predicted the same 5 variables as previously tested by the Plackett–Burman design. Table 4 reports the percentage decolorization with standard deviations for all tested conditions. The standard deviations were relatively low (from 0.1 to 7.7% in percentage units of decolorization). R² was 89.78%, R²(adj) was 87.85%, and RMSE was 10.48% (in percentage units of decolorization), thus indicating good agreement between experimental and predicted values using the proposed model.

The second-order polynomial equation, indicating the main effects, interactions among variables, and the quadratic effect with coefficients obtained from statistical analysis are represented as follows (Equation (5)):ŷ = 64.07 + 20.61⋅x_1_ − 20.69⋅x_2_ + 4.69⋅x_3_ − 21.10⋅x_4_ + 10.16⋅x_5_ − 18.57⋅x_1_² + 1.54⋅x_2_² − 11.72⋅x_3_² − 33.40⋅x_4_² − 3.60 x_5_² + 9.78⋅x_1_⋅x_2_ + 4.04⋅x_1_⋅x_3_ − 12.88⋅x_1_⋅x_4_ + 2.45⋅x_1_⋅x_5_ + 1.94⋅x_2_⋅x_3_ + 10.41⋅x_2_⋅x_4_ + 4.33⋅x_2_⋅x_5_ − 13.17⋅x_3_⋅x_4_ + 10.57⋅x_3_⋅x_5_ − 1.71⋅x_4_⋅x_5_
(5)

According to Equation (5), laccase-like activity (x_1_), initial dye concentration (x_2_), and pH (x_4_) were the most effective variables with the highest coefficients in a linear regression with positive (x_1_) and negative (x_2_, x_4_) effects. Moreover, as illustrated in Table 5, ANOVA on RB5 decolorization confirmed that *p*-values were less than 0.001 and 0.01 for linear regression (factors without their interactions), thus making all factors highly significant. Regarding the quadratic effect of factors, some of them were statistically non-significant with *p*-values over 0.05, such as initial dye concentration (*p* = 0.465) and temperature (*p* = 0.089), whereas the rest of the quadratic-term factors (laccase-like activity, HBT concentration, and pH) had a significant influence (*p* < 0.001). Regarding the interactions between the five variables, 5 among the 10 were significant. The highest *p*-values found were for pH × temperature (*p* = 0.592) and dye × HBT concentration (*p* = 0.561). pH × laccase-like activity and pH × HBT concentration were the most significant and valuable interactions (with the highest negative coefficients; Equation (5)) and had a significant influence on RB5 decolorization (*p* < 0.001). Moreover, the initial dye concentration × pH interaction was found to be significant with a positive-signed effect and a *p*-value < 0.01.

Residual plots (Figure 2) produced using the Box–Behnken design showed that all points were aligned on the normal line, confirming that the model used (Equation (5)) fits the experimental values (Figure 2A). This is also confirmed for the residual vs. fit and residual vs. order plots, which demonstrated that all experimental points were randomly distributed within the whole domain (Figure 2B,C). Figure 2D presenting the frequency of residuals clearly shows that the highest frequency (40%) was for the zero residuals. The second-highest frequency (at about 25%) was for the residual equal to 5 (in percentage units of decolorization), and all the remaining values of residuals had a frequency of less than 10%. Based on these findings, the adopted model showed a good fit.

Iso-responses plots (contour curves) were analyzed to visualize the interaction between factors. Figure 3A showing RB5 decolorization computed as a function of dye concentration and HBT concentration (all other factors were held at center level) indicates that the increase in dye concentration from 25 mg L^−1^ (level −1) to 125 mg L^−1^ (level +1) decreased RB5 decolorization from 80% to 30%. Increasing HBT concentration to 2.5 mM (center of the experimental domain—level 0) improved RB5 decolorization compared to the condition with 0.5 mM of HBT. Contour plots for the laccase-like activity× temperature interaction (Figure 3B) showed that increasing both factors improved decolorization percentages from 20% to more than 70%. Contour plots for the laccase-like activity × pH interaction (Figure 3C) showed that maximum percentage decolorization, at more than 75%, happened at levels +0.75 and ~−0.5, respectively. Moreover, percentage decolorization as a function of pH × temperature (Figure 3D) and pH × HBT concentration (Figure 3E) reached more than 68% in both cases. However, the lowest levels of factors for both the pH and dye concentration variables together are likely to exceed 85% RB5 decolorization (Figure 3F).

### 3.3. Conditions Optimization

Minitab 16 statistics software was used to identify the optimized conditions for RB5 decolorization. The special condition combining all 5 variables included laccase-like activity (0.5 U mL^−1^—high level (+1)), dye concentration (25 mg L^−1^—low level (−1)), HBT concentration (4.5 mM—high level (+1)), pH (4.2—close to level 0 (−0.203)), and temperature (55 °C—high level (+1)). The best values for each variable to obtain maximum decolorization as predicted by the Box–Behnken design are highlighted in red (Figure 4). The optimized conditions obtained here were adopted and repeated four times experimentally, reaching 82 ± 0.6% decolorization. 

### 3.4. Decolorization Process in 500 mL Volume

The optimized condition-set (laccase-like activity 0.5 U mL^−1^, dye concentration 25 mg L^−1^, HBT concentration 4.5 mM, pH 4.2, and temperature 55 °C) obtained via the Box–Behnken design was tested in a total reaction volume of 500 mL to test the conditions in larger volumes. Figure 5 shows the kinetics of RB5 decolorization as a function of incubation time, reaching 77.6 ± 0.4% decolorization within 2 h of incubation (73.3 ± 0.4% in 30-min time-steps), followed gradually by a slight increase in decolorization to reach 84.7 ± 0.6% in 6 h. At 24 h of the incubation period, decolorization reached 86.4 ± 0.4%. The RB5 color changed from dark blue to intense yellow (Appendix A) during the decolorization process.

### 3.5. UV-VIS Spectrum

Spectrum analysis showed that the maximum absorbance peak of RB5 was at 598 nm, which is in the visible-light region (Appendix A). Moreover, the absorbance of naphthalene and benzene rings was observed in the UV region at 254 and 310 nm, respectively. In test conditions containing HBT, there was a significant peak with a high absorbance value in the 280–310 nm range that was almost absent in the other conditions, which may indicate the maximum absorbance of the HBT mediator. Compared to RB5 incubated solely with the crude *C. gallica* laccase, the presence of HBT led to a 14% increase in decolorization (82% vs. 67%) accordingly a decrease in the peak absorbance of RB5 at 598 nm. Note that there was no decrease in absorbance when RB5 was incubated with HBT but without enzyme.

## 4. Discussion

Textile manufacturing encompasses a complex cluster of various different technologies and machinery [7] that ultimately produce vast amounts of colored clothing but use vast amounts of water [66,67]. Textile-industry wastewater is discharged with various contaminants such as dyes that are associated with toxicity and hazardous effects on both health and the environment. The Reactive Black 5 studied here is a well-known azo dye responsible for 70% of global demand [68]. RB5 has a well-defined chemical structure and has been widely used in studies directed toward fungi-driven decolorization of dyestuffs in textile manufacturing effluents [69,70]. Using a laccase-like active cell-free supernatant of the basidiomycete *C. gallica* to decolorize RB5 could therefore be an efficient solution to minimize RB5 concentrations in wastewater. The experimental design successfully performed here used a screening plan that came with five factors and indicated that 4 of the 5 factor variables tested were significant (laccase-like activity, dye concentration, HBT concentration, and pH). Moreover, for the fifth variable, which was temperature, we choose to rule it as significant as it was very close to the *p* = 0.05 limit (*p* = 0.073). Daâssi et al. (2012) [71] found that the same five factors as described here had significant effects on RB5 decolorization using a crude laccase from *T. trogii*. However, whereas the Pareto chart (Appendix A) indicated that laccase-like activity was the most effective factor for RB5 decolorization here, Daâssi et al., (2012) [71] found that with *T. trogii*, enzyme concentration had a minor effect on RB5 removal. Likewise, the decolorization of two reactive dyes, i.e., Reactive Blue 114 (RB114) and Reactive Red 239 (RR239), was unaffected by the enzyme concentration of a commercial laccase from *Aspergillus* sp. [72]. Furthermore, our results showed that pH was the second important factor after laccase-like activity. pH has been considered a relevant factor that mostly affects the decolorization reaction as it is a decisive factor for optimum activity and stability of the enzyme [73,74]. For instance, the optimum pH of the laccase of *C. gallica* is 3.0 (see Songulashvili et al. (2016) [75]) but the enzyme has only demonstrated stability in a range of pH 4.0 to 7.0 [37]. Our analysis of the pH effect found a negative slope that was very different from that found for the laccase-like activity factor. Here, the increase in pH (from 3 to 6) decreased decolorization efficiency. Dye removal by crude laccase from *Trametes* sp. was found to be sensitive to small pH changes [71]. When using the whole fungus culture for the decolorization of RB5 by *P. eryngii* F032, the optimum pH occurred at 3, affording a maximum of 94.56% dye removal, whereas an increase in pH up to 10 decreased RB5 decolorization down to just 25.9% [76]. However, for free and immobilized cells of *T. versicolor* and *Yarrowia lipolytica* NBRC 1658, increasing pH values increased the RB5 decolorization rates [77,78].

The Box–Behnken experimental design and RSM then served as a suitable approach to facilitate the process modeling and determine the optimized conditions for improving the decolorization rate. The contour plots indicated that increasing the initial dye concentration has a negative effect on RB5 decolorization. Our results are consistent with data from El Bouraie and El Din (2016) [79] and Khan et al. (2021) [80] demonstrating that a high concentration of dye progressively decreases the efficiency of color removal. In a similar way, Bonugli–Santos et al. (2016) [81] showed that increasing dye concentration to over 200 mg L^−1^ negatively affected RB5 decolorization by *Peniophora* sp. CBMAI 1063, possibly due to the inhibitory effect of the dye on laccase-like enzymes at high concentrations [82,83]. To increase the efficiency of the laccase treatment, redox mediators are often used [84]. Analysis of the contour curvature showed that the high concentration of HBT increased the percentage rate of decolorization. Daâssi et al. (2013 a) [28], Murugesan et al. (2007) [85] and Neifar et al. (2011) [86] all observed similar results. Despite the role it plays in improving decolorization rates, HBT may inhibit the enzyme at HBT concentrations over 5 mM [85]. HBT already has an inhibitory effect on the crude laccase from *T. trogii* and laccase from *Trametes* sp. strain CLBE55 at up to 1 mM, [27,87].

Increasing the temperature parameter improved dye decolorization up to 55 °C. This result is in good agreement with the optimum temperature profile of the *C. gallica* laccase, which showed a correlation between the increase in temperature and laccase activity in the range of 20–60 °C [75]. Using the commercial laccase of *Aspergillus* oryzae, Wang et al. (2011) [88] reported that changing the temperature from 20 °C to 50 °C had no effect on the decolorization rate, but no explanation was provided for these results. Fernandez et al. (2020) [74] reached a similar conclusion with a crude secretome of *Pleurotus sajor-caju*, where increasing the temperature led to high levels of dye decolorization, with a maximum found at 35–40 °C. However, a study using cauliflower (*Brassica oleracea*) bud peroxidase to remove dyes (Reactive Red 2, Reactive Black 5, Reactive Blue 4, Disperse Orange 25, and Disperse Black 9) found that the decolorization rate was maximal at 40 °C and decreased at temperatures above 70 °C, probably due to the thermal denaturation of the enzyme [89]. 

The optimized conditions obtained here using RSM for laccase-like activity, dye concentration, HBT concentration, pH, and temperature were 0.5 U mL^−1^, 25 mg L^−1^, 4.5 mM, 4.2, and 55 °C, respectively, yielding 82 ± 0.6% color removal after 2 h of incubation. This result was in the same range for the decolorization of Tubantin direct dye bath for pH (4) and enzyme concentration (0.6 U mL^−1^ of laccase-active cell-free supernatant from *Trametes* sp.) leading to 88% decolorization [87]. However, the reaction parameters were held at different values, i.e., 75 mg L^−1^ of Tubantin bath dye, 1 mM HBT, 30 °C, and 20 h [87]. The laccase-active cell-free supernatant of *Trichoderma asperellum* was also reported to decolorize 90% of 50 mg L^−1^ RB5 within 24 h in the presence of an HBT mediator [90]. Complete decolorization was achieved on 50 mg L^−1^ of RB5 within 24 h using purified NADH-dichlorophenol indophenol (NADH-DCIP) reductase and lignin peroxidase (LiP) obtained from *Sterigmatomyces halophilus* SSA-1575 [91]. In conclusion, the result obtained in this work can be considered very competitive compared to others, especially given that the process used is short (2 h) and does not require any purification steps.

Tests were run at a larger scale (500 mL reaction volume) to increase the total quantity of dyes with the best condition-set and achieved 77.6 ± 0.4% decolorization within 2 h and compared to controlled bioreactor processes. Mohorčič et al. (2004) [12] reported that the immobilized mycelium of *Bjerkandera adusta* removed 95% of the azo dye RB5 (0.2 g L^−1^) in a 5 L aerated stirred tank bioreactor within 20 days. Pavko (2011) [92] reported that the white-rot fungus *T. versicolor* was capable of decolorizing 97% of the synthetic dye Orange G in a 1.5 L bioreactor within 20 h. In a 3 L-volume bioreactor, *Pseudomonas putida* SKG-1 strain was able to decolorize 98% of the reactive orange 4 dye within 60 h [93]. This shows the value of our results obtained with a 2-h process in non-controlled flask conditions and the reproducibility of our results between the 1-mL lab-scale and the larger 500-mL scale-up.

The UV-visible spectrum of RB5, in the range of 200 to 800 nm, indicated peak absorbance of RB5 at 598 nm together with the presence of benzene and naphthalene ring signatures at 254 and 310 nm, respectively, [94,95]. Spectra analysis showed that the addition of HBT decreased the absorbance peak of RB5 dye only in the presence of the laccase-like active cell-free supernatant of *C. gallica*, demonstrating that the laccase mediator system efficiently decolorizes RB5. Our results confirmed those of Tavares et al. (2008) [96] who showed that the presence of the 2,2′-azino-bis(3-ethylbenzothiazoline-6-sulphonique) mediator reduced the maximum absorbance peak of the dye and that the decolorization occurred when both the mediator and the enzyme were added to the reaction mixture.

## 5. Conclusions

The statistical design-of-experiments approach employed here showed that 4 factors among the 5 studied were key factors for the degradation of the RB5 dye. Among all these parameters, the main factor was the concentration of laccase activity used in the process. This is a critical point, as the price of the enzyme dictates the economic viability of the process. The second most critical factor was pH, probably due to the biochemical properties of the enzymes (pH optimum and pH stability). However, the present study confirmed that the laccase cell-free supernatant of *C. gallica* was active on RB5 dye using the mediator HBT, and that an enzymatic treatment of 2 h was sufficient to achieve up to 82 ± 0.6% decolorization in a lab-scale reaction volume of 500 mL Further scale-up experiments are required to demonstrate that this treatment can be made amenable to pilot scale and to future industrial applications. This work provides proof-of-concept for the biodegradation of textile-industry dyes as a promising alternative route to physicochemical treatments. Further experiments could usefully measure the biochemical oxygen demand (BOD) and chemical oxygen demand (COD) of the treated dye solution to evaluate the toxicity of the residual dye. Processes using cheaper and eco-friendly alternative plant-origin mediators should be tested in an effort to engineer a more sustainable system.

## Figures and Tables

**Figure 1 microorganisms-10-01137-f001:**
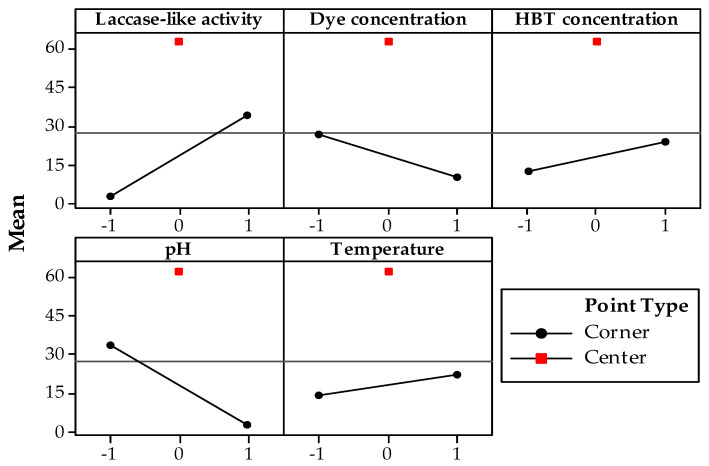
Main effects plot for RB5 decolorization after 120 min of reaction as function of coded value of factors. Red squares are center points and black diamonds are corner points.

**Figure 2 microorganisms-10-01137-f002:**
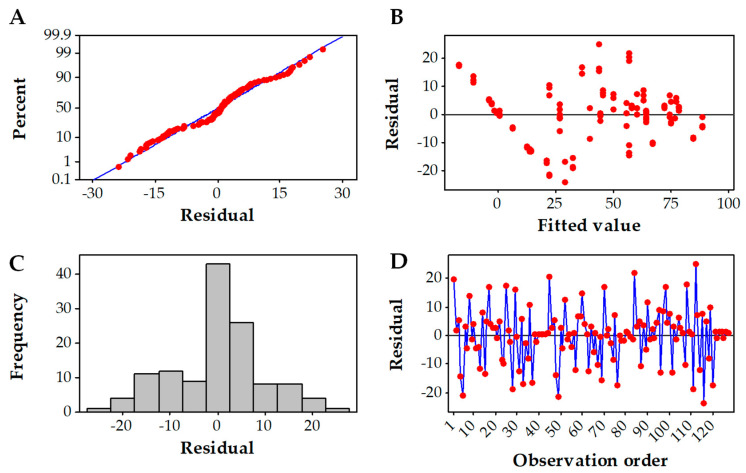
Residual plots for RB5 decolorization using the Box–Behnken design. (**A**) Normal probability plot. (**B**) Versus fits. (**C**) Histogram. (**D**) Versus order.

**Figure 3 microorganisms-10-01137-f003:**
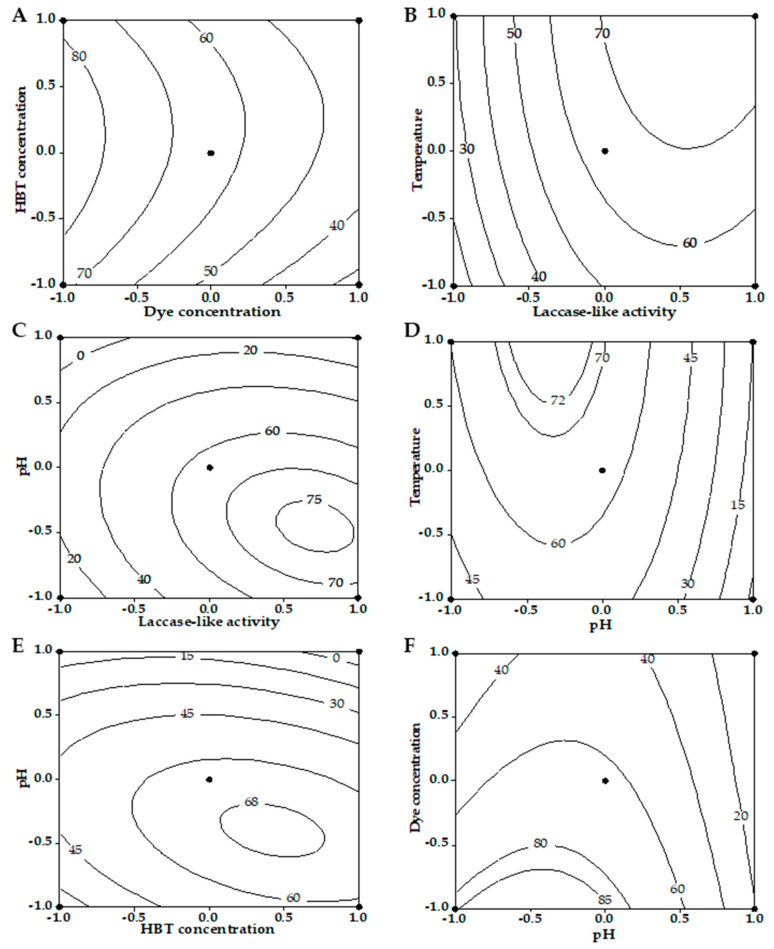
Iso-responses plots of RB5 decolorization as function of interactions between factors. (**A**) Dye concentration × HBT concentration interaction. (**B**) Laccase-like activity × Temperature interaction. (**C**) Laccase-like activity × pH interaction. (**D**) pH × Temperature interaction. (**E**) HBT concentration × pH interaction. (**F**) pH × Dye concentration interaction. All other factors were held at center level for each case.

**Figure 4 microorganisms-10-01137-f004:**
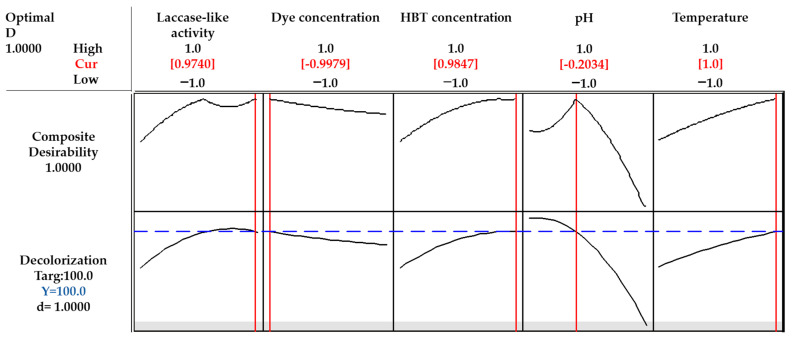
Optimized condition-set for RB5 decolorization obtained with Minitab 16 statistical software.

**Figure 5 microorganisms-10-01137-f005:**
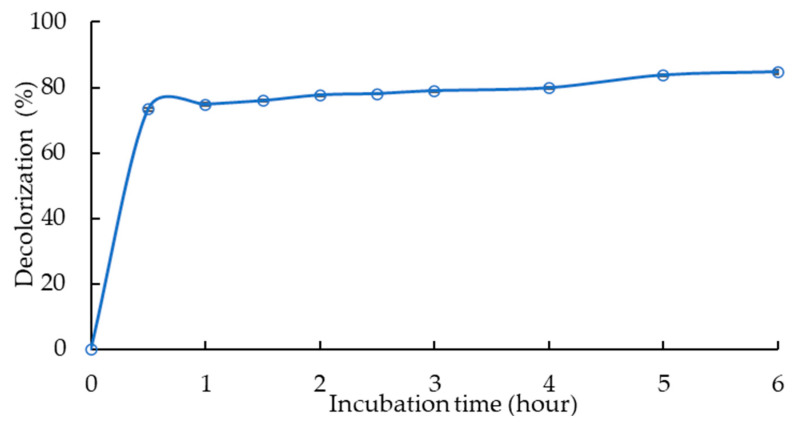
Kinetics of RB5 decolorization as a function of incubation time.

**Table 1 microorganisms-10-01137-t001:** Properties of Reactive Black 5 dye.

Properties	Reactive Black 5
CAS number	17095-24-8
Molecular weight (gmol^−1^)	991.82
EC Number	241-164-5
CI	20505
Dye content	≥50%
Chemical formula	C_26_H_21_N_5_Na_4_O_19_S_6_
λ_max_ (nm)	598
Molecular structure	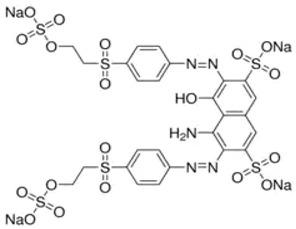

**Table 2 microorganisms-10-01137-t002:** Levels of factors tested in the screening design.

Symbol Code	Factor	Unit	Coded and Uncoded Levels		
			−1	0	+1
x_1_	Laccase-like activity	U mL^−1^	0.02	0.26	0.5
x_2_	Initial dye concentration	mg L^−1^	25	75	125
x_3_	HBT concentration	mM	0.5	2.5	4.5
x_4_	pH	-	3.0	4.5	6.0
x_5_	Temperature	°C	25	40	55

**Table 3 microorganisms-10-01137-t003:** Plackett–Burman screening-plan runs and response as percent decolorization at 120 min.

Run	Symbol Code of Factors					Decolorization (%)
	x_1_	x_2_	x_3_	x_4_	x_5_	
1	+1	−1	+1	−1	−1	74.8 ± 0.4
2	+1	+1	−1	+1	−1	1.0 ± 0.2
3	−1	+1	+1	−1	+1	1.3 ± 0.6
4	+1	−1	+1	+1	−1	5.2 ± 0.3
5	+1	+1	−1	+1	+1	1.3 ± 0.6
6	+1	+1	+1	−1	+1	55.8 ± 1.2
7	−1	+1	+1	+1	−1	1.3 ± 0.4
8	−1	−1	+1	+1	+1	6.0 ± 0.7
9	−1	−1	−1	+1	+1	3.1 ± 0.1
10	+1	−1	−1	−1	+1	65.7 ± 3.2
11	−1	+1	−1	−1	−1	1.2 ± 0.4
12	−1	−1	−1	−1	−1	3.7 ± 1.1
13	0	0	0	0	0	62.0 ± 0.2
14	0	0	0	0	0	62.3 ± 0.3
15	0	0	0	0	0	63.4 ± 0.2

x_1_, x_2,_ x_3,_ x_4,_ x_5:_ coded variables for laccase-like activity, initial dye concentration, HBT concentration, pH, and temperature, respectively.

**Table 4 microorganisms-10-01137-t004:** The Box–Behnken design and data on percent decolorization at 120 min.

Symbol Code of Factors						Decolorization (%)
Run	x_1_	x_2_	x_3_	x_4_	x_5_	
1	−1	−1	0	0	0	77.6 ± 1.3
2	+1	−1	0	0	0	80.9 ± 0.7
3	−1	+1	0	0	0	1.3 ± 0.3
4	+1	+1	0	0	0	43.7 ± 1.8
5	0	0	−1	−1	0	0.8 ± 0.3
6	0	0	+1	−1	0	60.9 ± 0.3
7	0	0	−1	+1	0	1.7 ± 0.2
8	0	0	+1	+1	0	2.0 ± 1.0
9	0	−1	0	0	−1	75.6 ± 0.1
10	0	+1	0	0	−1	29.0 ± 1.9
11	0	−1	0	0	+1	85.4 ± 2.1
12	0	+1	0	0	+1	56.2 ± 4.1
13	−1	0	−1	0	0	0.8 ± 0.3
14	+1	0	−1	0	0	53.5 ± 1.0
15	−1	0	+1	0	0	0.6 ± 0.1
16	+1	0	+1	0	0	69.9 ± 1.8
17	0	0	0	−1	−1	52.6 ± 1.3
18	0	0	0	+1	−1	1.6 ± 0.3
19	0	0	0	−1	+1	63.7 ± 3.7
20	0	0	0	+1	+1	1.5 ± 0.2
21	0	−1	−1	0	0	74.7 ± 0.4
22	0	+1	−1	0	0	23.9 ± 2.8
23	0	−1	+1	0	0	82.8±0.8
24	0	+1	+1	0	0	37.0 ± 7.7
25	−1	0	0	−1	0	0.6 ± 0.0
26	+1	0	0	−1	0	56.6 ± 0.3
27	−1	0	0	+1	0	0.9 ± 0.2
28	+1	0	0	+1	0	0.3 ± 0.1
29	0	0	−1	0	−1	43.8 ± 1.5
30	0	0	+1	0	−1	15.0 ± 1.8
31	0	0	−1	0	+1	62.9 ± 5.1
32	0	0	+1	0	+1	76.4 ± 4.3
33	−1	0	0	0	−1	1.4 ± 0.2
34	+1	0	0	0	−1	55.1 ± 2.9
35	−1	0	0	0	+1	8.6 ± 5.0
36	+1	0	0	0	+1	74.9 ± 4.4
37	0	−1	0	−1	0	76.2 ± 0.2
38	0	+1	0	−1	0	31.5 ± 1.9
39	0	−1	0	+1	0	4.3 ± 0.5
40	0	+1	0	+1	0	0.8 ± 0.3
41	0	0	0	0	0	62.3 ± 0.7
42	0	0	0	0	0	64.0 ± 1.7
43	0	0	0	0	0	65.2 ± 0.4
44	0	0	0	0	0	64.2 ± 1.0
45	0	0	0	0	0	64.5 ± 1.0
46	0	0	0	0	0	64.2 ± 1.4

x_1_, x_2,_ x_3,_ x_4,_ x_5:_ coded variables for laccase-like activity, initial dye concentration, HBT concentration, pH, and temperature, respectively.

**Table 5 microorganisms-10-01137-t005:** Analysis of variance for RB5 decolorization.

Source	Degree of Freedom	Sum of Squares	Mean Square	F-Value	*p*-Value Probability > F
Regression	20	102,350	5117.5	46.6	<0.001
x_1_	1	17,064	17,063.8	155.2	<0.001
x_2_	1	19,526	19,525.6	177.7	<0.001
x_3_	1	931	931.0	8.5	0.004
x_4_	1	19,003	19,003.2	172.9	<0.001
x_5_	1	4585	4584.6	41.7	<0.001
x_1_²	1	8115	8115.4	73.8	<0.001
x_2_²	1	59	59.1	0.5	0.465
x_3_²	1	3340	3339.9	30.4	<0.001
x_4_²	1	27,057	27,056.6	246.2	<0.001
x_5_²	1	323	323.3	2.9	0.089
x_1_ × x_2_	1	1148	1148.1	10.5	0.002
x_1_ × x_3_	1	162	161.6	1.5	0.228
x_1_ × x_4_	1	1477	1477.5	13.4	<0.001
x_1_ × x_5_	1	59	59.4	0.5	0.464
x_2_ × x_3_	1	37	37.4	0.3	0.561
x_2_ × x_4_	1	1300	1300.1	11.8	0.001
x_2_ × x_5_	1	225	225.4	2.1	0.155
x_3_ × x_4_	1	1891	1890.7	17.2	<0.001
x_3_ × x_5_	1	1341	1340.9	12.2	0.001
x_4_ × x_5_	1	32	31.8	0.3	0.592
Residual Error	106	11,649	109.9		
Total	126	114,000

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
