# Peer review of "Optimization of the Decolorization of the Reactive Black 5 by a Laccase-like Active Cell-Free Supernatant from Coriolopsis gallica"

_microorganisms, 2022, doi:10.3390/microorganisms10061137_

Round 1
Reviewer 1 Report
The quality of Figure 1. Shift of RB5...., must be improved.
The quality of molecular structure in Table 1 must be improved.
Please check correct number of significance firgures in table 6.
Author Response
We would like to thank Reviewer 1 for his comments and you will find our answers.
Comment 1: The quality of Figure 1. Shift of RB5...., must be improved.
Response 1: In fact, shift of RB5 corresponded to Figure S1 and the quality of this figure was improved as requested. In addition, Figure 1 (Pareto Chart of the standardized effects for RB5 decolorization) was moved to supplementary data as Figure S2 (as suggested by reviewer 2).
Comment 2: The quality of molecular structure in Table 1 must be improved.
Response 2: The quality of molecular structure in Table1 has been improved.
Comment 3: Please check correct number of significance figures in table 6.
Response 3: Number of significance figures was corrected as suggested.
Reviewer 2 Report
The present manuscript describes the optimization of Reactive Black 5 dye by a crude enzyme preparation from Coriolopsis gallica, containing laccase activity. The manuscript is very well written, carefully designed and technically accurate. Therefore, only minor changes are required for publication. However, it lacks novelty, and this is also indicated by the reference list, where most of the cited papers are more than 5 years old. The fact that white-rot fungal laccases are able to decolorize reactive dyes is known for several years, and I do not see what this study has to offer in the existing literature.
My minor comments are the following:
L84: Rhus vernicifera is not a fungal strain, it is a plant.
L99-101: This sentence does not make sense, please rephrase.
L120: The strain has to be deposited in a publicly accessible culture collection, and the relevant information (link to the database, strain accession number) have to be disclosed in the manuscript, in order for the experiments to be reproducible by an independent lab.
L133-135: It does not make sense to add the laccase inducer on the same day that the culture supernatant was collected. This is not mentioned in the cited ref 65 either.
Perhaps Table 3 and 4 should be added to the Results section, because they contain experimental results. Also, there are a lot of tables in the manuscript, perhaps some of them should be transferred to supplementary material.
Equation 3 should be formatted correctly, the (3) is written twice.
L226: Please explain for how long the laccase was stable in acidic pH in the previous study cited.
Equation 4 and Figure 1 are redundant. Figure 1 can be moved to supplementary material.
All figures should be provided in a significantly improved resolution. The diagrams of Figure 4 should be replaced with contour plots.
L460-461: Not all factors were significant, please revise.
Author Response
We would like to thank Reviewer 2 for his comments and you will find our answers.
Comment 1: L84: Rhus vernicifera is not a fungal strain, it is a plant.
Response 1: L84: “Rhus vernicifera” was deleted and replaced by reference 35.
- Kiran S, Huma T, Jalal F, Farooq T, Hameed A, Gulzar T, Bashir A, Rahmat M, Rahmat R, Rafique MA. Lignin degrading system of Phanerochaete chrysosporium and its exploitation for degradation of synthetic dyes wastewater. Polish J Environ Stud, 2019, 28(3), 1749-1757. DOI: https://doi.org/10.15244/pjoes/89575
In addition, we changed reference 34 by a more recent publication of 2020.
- Xu L, Sun K, Wang F, Zhao L, Hu J, Ma H, Ding Z. Laccase production by Trametes versicolor in solid-state fermentation using tea residues as substrate and its application in dye decolorization. J Environ Manage, 2020,270, 110904. https://doi.org/10.1016/j.jenvman.2020.110904
Comment 2: L99-101: This sentence does not make sense, please rephrase.
Response 2: L99-101: The sentence was replaced as follows: However, laccases are unable to directly oxidize complex compounds that have a higher redox potential than laccases or steric hindrance preventing access to the catalytic site of the enzyme.
Comment 3: L120: The strain has to be deposited in a publicly accessible culture collection, and the relevant information (link to the database, strain accession number) have to be disclosed in the manuscript, in order for the experiments to be reproducible by an independent lab.
Response 3: L120-122: Coriolopsis gallica strain CLBE 55 was deposited at the culture collection “Centre International de Ressources Microbiennes”(CIRM) under accession number BRFM 3473.
Comment 4: L133-135: It does not make sense to add the laccase inducer on the same day that the culture supernatant was collected. This is not mentioned in the cited ref 65 either.
Response 4: L133-135: We would like to think Reviewer 2 for his comment as it was typing error. The culture was grown for 7 days instead of 3 days. The sentence was corrected as follows:
“pH was adjusted to 5.5, and cultures were incubated in a rotary shaker for 7 days at 30°C and 160 rpm. On day 3 of incubation, 300 µM CuSO4 were added as a laccase inducer.
Reference 65 “Zouari-Mechichi H, Mechichi T, Dhouib A, Sayadi S, Martinez AT, Martinez MJ. Laccase purification and characterization from Trametes trogii isolated in Tunisia: decolorization of textile dyes by the purified enzyme. Enzyme Microb Technol, 2006, 39(1), 141-148. https://doi.org/10.1016/j.enzmictec.2005.11.027 , was deleted.
Comment 5: Perhaps Table 3 and 4 should be added to the Results section, because they contain experimental results. Also, there are a lot of tables in the manuscript, perhaps some of them should be transferred to supplementary material.
Response 5: Table 3 and 4 were included in the results section.
Table 5: “Decolorization conditions” was moved to supplementary material (Table S1).
Comment 6: Equation 3 should be formatted correctly, the (3) is written twice.
Response 6: The second (3) has been deleted.
Comment 7: L226: Please explain for how long the laccase was stable in acidic pH in the previous study cited.
Response 7: As suggested we have added this item in the text: “In addition, a previous study had demonstrated that the main laccase of C. gallica was active and stable at acidic pH (4-7) for 24 hours of incubation retaining from 40 to 80% of its activity”.
Comment 8: Equation 4 and Figure 1 are redundant. Figure 1 can be moved to supplementary material.
Response 8: Figure 1 was moved to supplementary material and then it became Figure S1 (see answer for Reviewer 1).
Figure S1. Pareto Chart of the standardized effects for RB5 decolorization (after 120 min of reaction) as function of coded value of factors.
Comment 9: All figures should be provided in a significantly improved resolution. The diagrams of Figure 4 should be replaced with contour plots.
Response 9: All figures were significantly improved for resolution and their order of appearance was changed.
In the corrected manuscript, Figure 4 became Figure 3 and was replaced with contour plots and the results were changed accordingly: lines 356-370.
Comment 10: L460-461: Not all factors were significant, please revise.
Response 10: The sentence was revised as follows: The statistical design-of-experiments approach employed here showed that 4 factors among 5 studied were key factors for the degradation of the RB5 dye.